# Cultural Differences in Patients’ Preferences for Paternalism: Comparing Mexican and American Patients’ Preferences for and Experiences with Physician Paternalism and Patient Autonomy

**DOI:** 10.3390/ijerph191710663

**Published:** 2022-08-26

**Authors:** Gregory A. Thompson, Jonathan Segura, Dianne Cruz, Cassie Arnita, Leeann H. Whiffen

**Affiliations:** Department of Anthropology, Brigham Young University, 800 KMBL, Provo, UT 84602, USA

**Keywords:** paternalism, patient autonomy, healthcare, culture, preference, practices, physician-patient interaction, White American, Mexican American, Mexican

## Abstract

Following up on previous research demonstrating the high level of care realized by a paternalistic Mexican physician, the present research further explored the hypothesis that there are cultural differences in preferences for and experiences with physician paternalism vs. patient autonomy in White American culture as compared with Mexican culture. In this research, we interviewed sixty (60) people including twenty (20) Mexican, twenty (20) Mexican American, and twenty (20) White American respondents. We asked these patients about their experiences with and attitudes towards paternalism and patient autonomy in healthcare interactions. With some caveats, our data showed strong support for both hypotheses while also suggesting a high level of care can be realized by paternalistic physicians when “paternalism” is understood in a cultural context. We close with a brief consideration of the implications of these findings.

## 1. Introduction

### Is Autonomy a Cultural Universal?

Autonomy in healthcare has been widely accepted in Western medicine as one of the most important factors in bioethics [1,2,3,4,5,6,7,8]. Beauchamp and Childress included “Respect for autonomy” as one of their four key principles of biomedical ethics [9,10]. Beauchamp and Childress define the autonomous individual as someone who “acts freely in accordance with a self-chosen plan, analogous to the way an independent government manages its territories and sets its policies” [9] (p. 58). Although Beauchamp and Childress themselves argue that this definition of autonomy means that a patient choosing a paternalistic physician could be ethical even if it means a relative loss of that patient’s autonomy in the physician-patient interaction, there has been some criticism of autonomy from within Western cultural contexts (e.g., most Western researchers and writers have considered paternalism in physician-patient interactions to be inimical to providing high quality care for patients [11,12,13,14]).

Respect for patient autonomy has also been central to principles and practices in healthcare in the West. For example, the Belmont Report (1979) names the “respect for persons” or autonomy, as its first basic ethical principle [15]. Article 5 in the UNESCO Universal Declaration on Bioethics and Human Rights (2009) states that: “The autonomy of persons to make decisions, while taking responsibility for those decisions and respecting the autonomy of others, is to be respected” [16] (p. 7).

Autonomy has been taken to be of such importance that it has often been thought to be ethically necessary to introduce it to other cultures where it is not highly valued. For example, while agreeing with Thompson and Whiffen’s (2018) basic findings regarding a tendency towards paternalism in physician-patient interactions in Mexican culture [17], Lazcano-Ponce et al. (2020) argue that paternalism is equal to overprotection and they further argue that Mexican physicians need to more forcefully be “promoting autonomy in the doctor-patient relationship” [18] (p. 9). Yet, not everyone has been so sanguine about this kind of ethical imperialism. In arguing for a pluralistic view of ethical values, Shweder (2004) has suggested that it would be better to consider (at least) three ethics that can be taken as a preeminent value in a given culture and which can often contradict one another [19]. Autonomy is just one of these three ethics; community and divinity are the other two. As Jensen (2011) describes the pluralism of this approach: “The model, then, is not a simple one-size-fits-all, but rather accommodates the prevailing ethics of diverse peoples” [20] (p. 157). Many have noted that whereas Western culture values autonomy over both community and divinity, other cultures may place community or divinity as preeminent values in such a manner that those values can supersede autonomy. For our present purposes, this suggests that the preeminent ethical principle of autonomy in healthcare may not be a preeminent value in other cultures and that there may be other values that supersede patient autonomy [21,22,23,24,25,26,27,28].

In a similar vein, others have argued that because most research regarding physician-patient interactions has been conducted and interpreted from a Western perspective, paternalistic healthcare practices are regularly seen as negative even in cultures where paternalism is seen very positively [23,28,29]. A number of researchers have provided strong evidence that a paternalistic approach to healthcare is preferred in other countries. This research has included countries such as Ghana [29,30], Botswana [31], Ethiopia [32], Jamaica [33], Korea [34,35], China [36], India [27,37], Vietnam [28], Japan [38], Thailand [39,40], Pakistan [41], Turkey [42], Southern European Nations [43], and Mexico [17,24]. Others have pointed to the need to understand how physician-patient interactions are inflected by the values and ideals of the cultural contexts in which those interactions take place [21,28,30,31,34,44,45,46].

Taking just a few examples of how patient autonomy is often superseded by other values, Sousa (2011) shows how in India, physicians take the family’s wishes into account, even when they may conflict with the patient’s wishes [27]. Veatch (1985) documents a similar emphasis in China where it is common practice to inform the patients’ families before speaking with the patient [46]. Ujewe (2018) and Payne-Jackson et al. (2004) have separately shown that there are similar ethical principles emphasizing community over autonomy in patient’s interactions with healthcare providers in Africa and Jamaica [30,47]. Norman (2015) documented how Sub-Saharan African patients expect physicians to tell them what is wrong and that paternalism actually enhances the health seeking behavior of patients [29].

In direct contrast to those arguing for the imposition of autonomy in other cultures, others have even argued that applying an autonomous approach in healthcare in non-Western cultures leads to poor patient care [31,34,45]. In Botswana, Shaibu (2007) explains that autonomous decision-making in healthcare is not correlated to quality care and that the Western model contributes to “incongruence between the cultural values of Batswana (people of Botswana) and those of health workers as modeled on the western ethic” [31] (p. 503). Similarly, Betancourt (2000) relates the story of Mrs. Y, a sixty-year-old Japanese American woman, who visited an emergency room in the U.S. for a fever and bruises. When Mrs. Y’s family were told her diagnosis of leukemia, they insisted that they had the right to withhold this diagnosis from Mrs. Y (in spite of the family’s wishes, the American doctors told Mrs. Y her diagnosis, feeling that it was essential to let her make her own decisions about her care) [21]. Lai (1995) offers a similar example from Chinese culture where an autonomous approach disrupted emotional harmony by conflicting with the Confucian concept of *hsiao* [48]. McGrath et al. (2001) showed that participants from Indian, Filipino, Chinese, and Italian families described “The Western Way of informing people directly [about the terminal nature of an illness] was “too abrupt”, “terrifying”, and “blunt”” [44] (p. 307). Indeed, even some research from within Western contexts has suggested similar limits on the ethic of patient autonomy. Glaser and Strauss (1979) and Timmermans (1994) have separately demonstrated how the autonomy of dying patients can be limited by the awareness that is afforded them of their condition by others [49,50].

Taken together, this body of research offers strong evidence that patient autonomy is not necessarily the preeminent value in many healthcare settings around the world. Research on healthcare in Mexico suggests that Mexican healthcare is one of those cultural contexts in which physician paternalism may supersede patient autonomy.

Indeed, a number of researchers have shown that paternalistic practices are common in Mexican healthcare [51,52,53,54,55]. Garcia-Gonzalez (2009) found that the majority of patients who had chronic rheumatic diseases indicated they wanted to play a passive role and have the physicians make decisions for them [56]. Álvarez Bermudez (2018) analyzed interactions between nurses in Mexico and patients who had been hospitalized for more than three days and found that although less than half of those patients reported the nurses listening to their opinion, almost 70% of the patients characterized the attitude and communication of the nurses as “good” [57]. In a survey focusing on public perception and experiences of the Mexican healthcare system, Doubova et al. (2016) found almost 80% of their 1181 Mexican respondents reported receiving good quality care with 79.6% reported the primary care provider explains things in a way that is easy to understand; and 81% reported a primary care provider who solves most of the health problems [58] (p. 839).

Based on observations comparing over 40 physician-patient interactions of a Mexican physician with those of a White American physician, Thompson and Whiffen (2018) found that whereas the White American physician oriented to patient autonomy, the Mexican physician was paternalistic [17]. Importantly, they noted that far from appearing overbearing or domineering, the Mexican physician’s paternalism was realized in a way that his Mexican patients appeared to appreciate as demonstrating a strong sense of care for his patients. These findings suggested that paternalistic physicians can demonstrate high quality care in Mexican culture. Thompson and Whiffen (2018) based this conclusion on observations of Mexican patients’ reactions to their physicians [17]. Notably, due to IRB restriction and time challenges, their study lacked interview data with Mexican patients. The present research seeks to address this lack through interviews with sixty (60) participants from Mexican, White American, and Mexican American backgrounds regarding their preferences for as well as their experiences with patient autonomy and physician paternalism.

## 2. Methods

### 2.1. Participants

We completed a total of sixty (60) interviews with twenty (20) of the interviews coming from each of the following three groups: White American (note: We use the terms “White American”, “Mexican”, and “Mexican American” advisedly, recognizing the complex and potentially problematic nature of these terms. Both “White American” and “Mexican” are peculiar categories. Whereas the former is a racial category, the latter is a nation-state based category. Moreover, both categories have substantial intra-group diversity. Nonetheless, as we show in what follows, these problematic categorical boundaries correspond with substantial differences in preferences and thinking about autonomy and paternalism in relation to healthcare encounters. These differences are what we call “cultural differences”. In doing so, we caution the reader to keep in mind intracultural variation so as not to overly reify or essentialize these “cultural” categories.), Mexican American, or Mexican. White American participants were U.S. citizens who self-identified as “White”. Mexican American participants were born either in the U.S. and have parents or grandparents from Mexico or were born in Mexico. All Mexican American participants currently reside in the U.S. but citizenship status remained unidentified in case revealing that status might lead to harm (e.g., deportation). Mexican participants were born in Mexico and currently reside in Mexico. The participants came from several states in the U.S. and Mexico (see Table 1). Informed consent was obtained from all subjects involved in the study. The study was approved by the Institutional Review Board of Brigham Young University (approved 9 January 2021).

Participants were identified based on snowball sampling that began with researchers’ acquaintances. 59 of the 60 interviews were conducted over Zoom. One interview was conducted in person due to their proximity to the interviewer. Interviews were conducted in the language of the participant’s choice which was determined in the initial contact based on the participants’ preference. All interviews with White American respondents were conducted in English; all interviews with Mexican respondents were conducted in Spanish, and 17 interviews with Mexican American respondents were conducted in English with only three conducted in Spanish. Due to IRB concerns with the immigration status of participants, we do not have precise data for Mexican American participants’ length of time in the U.S. or of which generation they are. Nonetheless, the high proportion of Mexican American participants who chose English as their preferred language (17/20) suggests that these participants have been in the U.S. long enough to have substantially acculturated to mainstream American culture. This and other participant information including age, ratio of female-male, and interview duration is included in Table 2 below.

### 2.2. Interview Questions

We had a basic interview protocol that we aimed to ask each participant (See Appendix A), but, due to time constraints, some less important questions needed to be skipped with some respondents. Most questions were open-ended questions seeking to assess participant’s views, opinions, and experiences regarding patient autonomy and physician paternalism in their interactions with physicians, whether in the U.S. (for White American and Mexican American respondents), or Mexico (for Mexican and Mexican American respondents). Interviewers followed up these questions with unscripted probing questions to explore respondents’ reasoning. These typically included questions such as: ‘why do you prefer that?’, ‘can you give me an example?’, or ‘do you have good or bad experiences regarding that?’.

Regarding paternalism and autonomy specifically, we asked two main questions (see Questions 20 and 21 in Appendix A). Question 20 asks about their preferences for either paternalism or patient autonomy and 21 asks which one of these their physician most closely practices. Before asking any questions about paternalism and autonomy, we provided the following simple definitions of autonomy and paternalism:

Paternalism/*El Paternalismo:* A relationship in which the physician has substantial/most control and authority over the patient’s healthcare decisions./*Una relación en la que el médico tiene un control sustancial/ mayor y más autoridad sobre las decisiones de atención médica del paciente*.Autonomy/*La Autonomía*: A relationship in which the patient has substantial/most control and authority over their own healthcare decisions./*Una relación en la que el paciente tiene un control sustancial/ mayor y más autoridad sobre sus propias decisiones de atención médica*.

With these and other questions, we did not specify the context or the kind of physician. Rather, we left this up to the participants to define. Most talked about their primary care providers but some mentioned specialists of various kinds.

In addition to exploring these central questions, we also included an untested and exploratory method which involved a set of eight (8) “hypothetical” statements that were actual statements said in the physician-patient interactions described in Thompson and Whiffen (2018). For each of these hypothetical statements, patients indicated how unusual or weird they thought that statement would be if their doctor were to say it. Although the trend of those responses was in support of the cultural difference hypothesis, we did not have sufficient measures in place to ensure that participants understood the hypotheticals sufficiently well. As a result, in what follows, we have focused on participants’ answers to and explanations of their answers to the central questions.

### 2.3. Methods of Analysis

The audio recordings of these interviews were transcribed using Otter.ai and Sonix.ai, and then corrections were made by a Research Assistant. These transcripts were then coded in MAXQDA (See Appendix B and Appendix C) based on basic principles of in vivo grounded theory.

We created two codebooks to organize the data. The first codebook (Appendix B) organized the data with basic codes such as: paternalism, autonomy, doctor authority, doctor professionalism, physical and emotional caring, moral advice, etc. Each of these codes was broken down into subcodes. For example, the code *Care1-Physical and emotional caring* was broken down into four subcodes: *Actions demonstrating lack of care*, *Interpretations of actions demonstrating lack of care*, *Actions demonstrating care 1*, and *Interpretation of actions demonstrating care 1*. The second codebook (Appendix C) aimed to separate and define patient autonomy and physician paternalism according to participants’ descriptions. This included codes such as: *Autonomy—informed decision making* and *Paternalism—doctor’s expertise*, as well as their preferences which were organized into *autonomy*, *paternalism*, or *mix*. Interviews were coded by each member of the research team, and each member of the research team checked the other member’s codes to ensure that researchers’ coding was consistent across researchers. Results relevant to the cultural difference hypothesis are described below.

## 3. Results

We begin with the quantitative data regarding respondents’ preferences for patient autonomy vs. physician paternalism. We then explore these data further by describing our respondents’ explanations of their answers to this question. In 3.2, we turn to respondents’ answers to the question of actual experiences with physicians, again beginning with the quantitative data and then adding nuance and detail with the qualitative data describing their explanations for their answers. Because it is difficult to know how acculturated Mexican Americans are to American culture, in order to best examine the cultural difference hypothesis, we will focus primarily on the comparison between the groups of White American and Mexican respondents when considering respondents’ preferences for patient autonomy vs. physician paternalism. Yet, since Mexican Americans have experience with both White American *and* Mexican physicians, we include their responses in our analysis of respondents’ experiences with their physicians.

One other caveat is in order. To be sure, our study is not intended as an hypothesis-testing study (which would require a more rigorous selection methodology and more rigorous statistical methods of analysis). Rather, our study is intended as an hypothesis-exploring study in as much as we explore the cultural difference hypothesis in light of our participants’ responses. Our findings should be considered as preliminary findings and, we hope, serve as the basis for future hypothesis-testing research. However, we also feel that these data are quite compelling as they are.

### 3.1. Group Preferences Data

Figure 1 shows the number of respondents in each group expressing a preference for patient autonomy (please note that throughout the result section when we refer to “autonomy”, we mean “patient autonomy”), paternalism (by which we mean “paternalism of the physician”), or a mix of both in their interactions with their physicians.

The cultural difference hypothesis that we explored in this study would suggest that White American respondents prefer “Autonomy” and the Mexican respondents prefer “Paternalism”.

Our quantitative results provide modest support for this hypothesis (see Figure 1). Although only five of eighteen Mexican respondents said that they preferred “Autonomy”, twelve of twenty of the White American respondents said that they preferred “Autonomy”. Similarly, whereas only one of the twenty White American respondents said they preferred “Paternalism”, six of the eighteen Mexican respondents said they preferred “Paternalism”. Notably, seven White American respondents and seven Mexican respondents indicated that their preference was some kind of mix of both, something that we will explore further below.

A simple Chi-squared analysis of the responses of White American vs. Mexican responses produces a *p*-value of 0.012, suggesting that this difference is statistically significant (at the *p* < 0.05 level). Yet, as we will see next, when we dig into respondents’ explanations of their answers to this question, the differences are even starker, especially for those who answered “Mix”.

#### 3.1.1. Group Preferences: White American Respondents

Beginning with White American respondents, 60% (12) said that they preferred “Autonomy”. The most common aspects of autonomy mentioned by White American respondents were: *Informed decision making, doctor and patient as collaborators,* and *control over their body.* Ten White American respondents mentioned informed decision making, sixteen mentioned doctor and patient as collaborators, and ten mentioned the patient having control over their body. In addition to these common aspects of their descriptions of autonomy, a common logic emerged among White American respondents regarding their justifications for paternalism. This logic is roughly as follows: (1) they respect their physician’s medical expertise, but (2) they are experts concerning their own bodies and (3) because bodily autonomy is paramount, they themselves should make the final treatment decision.

First, regarding their respect for their physician’s expertise, many White American respondents who preferred “Autonomy”, including Alyssa (30), Barbara (58), Megan (48), and Alexa (48), each noted the physician’s medical expertise (note: all respondent names included in this article are pseudonyms). This is captured well by Barbara’s comment: “The physician has the degrees, instruction, and knowledge on paper”. While respecting and recognizing the value of their physician’s medical training and expert knowledge, White American respondents also noted that this knowledge was not enough to justify locating the treatment decision with the physician.

Instead, White American respondents remarked that they are experts concerning their own bodies and that this knowledge is more important than the medical expertise of their physicians. As a result, these respondents felt that they themselves are the ones who should have the decision-making authority. Several White American respondents, including Irene (58), Julia (53), Alexa (48), and Jacob (32), each described the importance of their own knowledge regarding their bodies, summarized well by what Alexa said: “Because it’s my life and my body, and I have the right to make those decisions… They’re just looking at it from a medical standpoint”. Though the training and experience of the physician gives them some authority from a “medical standpoint”, many White American respondents pointed that it is “their life”, and “their body” and that this means that they should have the authority in the decision-making. Overall, White American respondents preferred being responsible for the final treatment decision.

Although “Autonomy” was the most common White American response, “Mix” was second with 35% (7) of White American respondents choosing “Mix”. In their explanations of their responses, these respondents offered explanations that were very similar to the logic of the White American respondents who preferred “Autonomy”. In their explanations of their decisions, Michelle (49), Emily (52), and Matthew (49) described a similar logic to those indicating a preference for “Autonomy”. They “ultimately” preferred the locus of control and decision-making authority to be the patient. As Emily shared: “I really do like the doctor to give me their opinion, but I think that ultimately, it’s the person’s life and they need to choose what’s best for them”. Each shared that the “ultimate” decision-maker is the patient. In justifying this response, they gave similar reasonings such as bodily autonomy, it’s the patient’s life, or the patient is the expert regarding their body. Each of these definitions and reasonings are very similar to the definitions and reasonings given by the American respondents who preferred autonomy.

As one last example, Susan (51), a White American respondent who said she preferred “Mix” but leaned towards paternalism also sounds similar to these White American respondents who preferred “Autonomy”. She shared: “I would want them [the physician] to be a little bit more in charge, but I would like to be a decision maker…I think that we have ultimate say in our own treatment”. Even though she said she leans toward paternalism, she explains that the patient still has the ultimate decision-making authority. Interestingly, she wants the physician to be “more in charge”, but ultimately, she wants the final say.

The reasonings and definitions behind “Mix” for these White American respondents who chose “Mix” as their preference clearly point towards a more autonomy-oriented relationship with the physician. This further supports the idea that the White American respondents prefer an autonomy-focused physician-patient relationship.

Of course, as seen in Figure 1, one White American respondent, Abby (40), stated that she preferred “Paternalism” noting: “[I’m] going because [I] have a problem… [I’m wasting my time] if I am not really going to do what they think is best for me, right?” Here Abby recognizes the medical experience and knowledge that the physicians have, yet she differs from the other White American respondents since she does not mention the importance of her knowledge of her body and life experience or of the importance of bodily autonomy. Instead, she leaves the final treatment decision with the physician. This offers an important reminder that the cultural difference hypothesis must not be *deterministic* in nature—as if culture necessarily determines behaviors. Rather, individual and intracultural variation must be acknowledged in any consideration of the role of culture in human behavior.

This deeper look into the White American respondents’ descriptions and explanations of their preferences suggests a different organization of the quantitative data. Of the seven White American respondents who chose “Mix” as their preference, five actually described a preference for “Autonomy” in their explanations, and two describe something closer to “Paternalism”. When considering their explanations for their answers, overall, seventeen White American respondents described a preference for “Autonomy” and three White American respondents described a preference that pointed more towards “Paternalism” (See Figure 2 below). This points towards a clear preference for patient autonomy among the White American respondents.

#### 3.1.2. Group Preferences: Mexican Respondents

As noted above, the most common preference described by Mexican respondents was “Mix” (39%, 7), with “Paternalism” being second (33%, 6), and “Autonomy” third (28%, 5). Although the distribution between these seems fairly even with only a slight preference for “Paternalism”, as we will show below, the qualitative data illustrates an even stronger tendency towards a preference for “Paternalism” since in explaining their choices, several respondents who chose “Mix” actually described “Paternalism”.

The overall logic of Mexican respondents’ reasoning shared some similarities with the logic of White American respondents but differed in important ways. This logic could be roughly described as follows: (1) they respect their physician’s medical expertise, (2) although they have some knowledge of their own life experiences and bodies (3) their physician should have the final say in treatment decisions. An additional dimension of their logic was that qualities like trust, confidence, and obedience were crucial for their understanding of physician paternalism.

Similar to the White American respondents, several Mexican respondents who answered “Paternalism” for their preference noted a need to respect the physicians’ expertise. Lucia (31), Diego (33), and Sara (43), each shared a sentiment that is well summarized by Josue’s (52) simple statement: “They are the experts”. These respondents described how the training, schooling, and experience of the physicians makes them the medical experts, something that the patient is decidedly not. When discussing their preferences for who makes the decision or if doctors should consult the patient, many Mexican respondents from those who preferred “Paternalism” or “Mix”, including Isabela (47), Rosalia (52), Diego (33), and Samantha (55), reiterated the doctor’s expertise such as what Samantha shared: “Pretending to take care of it without having the knowledge is impossible”. Similarly, Diego shared: “I feel that they are the ones that know… They have studied, they have the knowledge and experience necessary”.

The Mexican respondents’ descriptions point towards physicians’ knowledgeability as qualifying the physicians to be the decision-maker. Anthony (50), Isabela, Rosalia, Josue, and Valeria (55), among others, ultimately shared that the locus of control should therefore be maintained by the physician and the physician should be the decision-maker. This is summarized well by what Rosalia shared: “The doctor is the one who decides… because he is the one who knows”. Furthermore, importantly, Lucia (48), and Camila (39), specifically pointed out that the doctor’s training and experience gives each of them “confidence” in the physician as the decision-maker, as Lucia shared: “I trust that their study and their effort is really the best for me. I have confidence in my doctor regarding my health”.

Regarding the seven Mexican respondents who stated “Mix” as their preference, six of them tended to lean towards paternalism or describe paternalistic characteristics that were similar to the Mexican respondents who preferred paternalism. In defining their “Mix”, Camila, Yanira (40), Dexter (58), and Alvin (55), each shared how the physician is the one who knows and is the greater authority in making decisions. Yanira answered “Mix” as her preference, yet in her definition of what “Mix” means to her, she shared: “I like the part that the doctor gives you his indication and it is something that is not questionable. I understand that the doctor has to tell me what to do, but I also think that they could listen… to what I have done or how I feel”. Yanira clearly points to a paternalistic relationship as the doctor “has to tell [her] what to do”

Many of the Mexican respondents who said they preferred “Mix” actually described paternalism in their explanations. For example, Alvin, one of the Mexican respondents who preferred “Mix”, stated: “I’d rather them give me the treatment because they’re supposed to be the ones who know [best]”, then later shared, “[Mix], because if I finish my medicine then I don’t know what to get so I call the doctor and ask what does he suggest. They then tell me to buy this or that. So, that is when the decision is made between the both of us”. Alvin indicates that his contribution as a patient is simply to let the doctor know that his medication has run out. The doctor will then “tell me to buy this or that”. Here, Alvin clearly points to a paternalistic preference by his first statement even though he chose “Mix”.

Although both Yanira and Alvin preferred “Mix”, when specifically looking at the decision-making authority they clearly prefer paternalism. Other Mexican respondents who chose “Mix” as their preference later clarified that they leaned towards paternalism. These Mexican respondent’s preferences, including Diana (38) and Angelica (37), are summarized well by what Diana shared. After having answered “Mix” as her preference Diana then shared that the doctor has more influence “because I think the doctor knows his job well and knows what he is doing”.

In addition to these “Mix” responses that required some recoding, three of the five Mexican respondents who chose “Autonomy” as their preference actually described paternalism when explaining their preference. These respondents indicated that they have a choice whether or not to follow their physician’s orders but that it is the physician’s job to give orders and it is the patient’s responsibility to follow those orders if they want to be healed. For example, Bella (44), who stated her preference as “Autonomy”, shared: “I’m going to the doctor because I feel bad and I [want] him to help me. So, I have to do what he tells me if I want to feel good…In some way he’d have to explain it to me well so that I do it”. Bella clearly points to paternalism as she admits that she has to do what the doctor says if she wants to feel good, yet the doctor still has to explain it well for her to fully comply. Her preference for “Autonomy” tended to relate more to compliance with the treatment, which she said depended on how well the physician explains his decision rather than who makes the decision. Similarly, Ana, a Mexican respondent who chose “Autonomy”, said “there is always the security that [the] doctor has the ability and authority to recommend what is best for [the patient]”. Here, as with a third Mexican respondent, we see the respondent who chose “Autonomy” is actually describing a relationship that is “Paternalistic”.

In sum, six of the seven who answered “Mix” actually described paternalism and three of the five who answered “Autonomy” also emphasized paternalism. This means that when recounted according to their descriptions of their preferences, just three Mexican respondents preferred “Autonomy” while fifteen preferred “Paternalism” (See Figure 2).

When recounted to take into account what the respondents actually said (Figure 2), a strong trend emerges showing that whereas White American respondents strongly preferred autonomy, Mexican respondents strongly preferred paternalism. Although there were some important similarities in their explanations, such as the importance of the physician’s expertise as well as the importance of the patient’s knowledge of their own bodies, in the end these similar elements were interpreted quite differently. Whereas the White American respondents generally felt that the final treatment decision should be made by the patient due in large part to their rights, even responsibility, of bodily autonomy, the Mexican respondents felt that the physician should make the final treatment decision because their knowledge is far superior to that of the patient. The Mexican respondents also emphasized the importance of things such as the authority and trustworthiness of the physician, the obedience of the patient, and confidence in the physician.

### 3.2. Group Experiences Data

As will be shown below, the qualitative data on respondents’ descriptions of group experiences was initially somewhat less clear regarding differences between White American and Mexican respondents’ experiences. In particular, White American respondents’ answers to the question of whether their physicians are oriented to autonomy or paternalism appeared to contradict the cultural difference hypothesis. We will thus begin our consideration of this data on respondents’ experiences with physicians by presenting this apparently contradictory data.

#### 3.2.1. White American Respondents Experiences of Paternalism with U.S. Physicians

As we can see in Figure 3, when comparing White American and Mexican respondents’ answers to the question about whether the physicians that they have encountered are oriented to paternalism or autonomy, a very similar number of White American respondents viewed their doctors as paternalistic as did Mexican respondents (58% vs. 64%, or 79% vs. 82% if “Mix” is included).

This would appear to contradict the cultural difference hypothesis regarding respondents’ actual experiences with physicians which would instead have suggested that White American respondents would describe their physicians as oriented to autonomy and Mexican respondents would describe their physicians as oriented to paternalism. Although it is of course possible that White American physicians are paternalistic, Mexican American respondents’ answers suggest a different conclusion.

As noted above, we asked Mexican American respondents about their experiences with *both* American physicians and Mexican physicians. Considering that many of them had experience with physicians in both places, their responses provide the opportunity to directly compare experiences with Mexican and American physicians (see Figure 4 and Figure 5).

Figure 4 shows Mexican American respondents’ responses to this question regarding Mexican physicians as compared to Mexican respondents’ responses with their Mexican physicians. The distribution between these two groups’ responses is very similar with both groups describing Mexican physicians as paternalistic. Figure 5 shows Mexican American respondents’ responses regarding American physicians as compared to White American respondents’ responses with their American physicians. Here, the two groups differ rather significantly. Whereas the White American respondents tended to characterize American physicians as paternalistic, the Mexican American respondents tended to characterize their American physicians as autonomy-oriented. This of course raises an important question as to why these two groups characterize American physicians so differently?

#### 3.2.2. Qualitative Data Explaining Respondents’ Definitions of Paternalism

Respondents’ explanations can help answer this question. As just noted, Mexican American respondents are particularly important in this regard since they had experience with both American and Mexican physicians and can provide a direct comparison. We will begin with their explanations.

##### Mexican American Respondents’ Experiences with American and Mexican Physicians

In explaining their responses, several Mexican American respondents described the differences they saw between the physicians in the U.S. and Mexico. Mexican American respondents Victoria (37), and Andres (55), each said that in the U.S., doctors let patients decide what is best for them and give suggestions, but in Mexico it is “more obligatory” to follow the physician’s treatment decision. Victoria pointed out that while a U.S. physician asks the patient if they agree with the given recommendation, a physician in Mexico gives more of a treatment directive leaving little room for the patient to contemplate whether or not to comply. Andres describes a very similar situation to Victoria, stating that Mexican physicians are much more insistent and authoritative by often making decisions before even asking the patient. Interestingly, Andres also notes that the authoritative physician who says “you have to do this” is “like a parent”. Andres’ and Victoria’s descriptions highlight what many Mexican American respondents described as the differences between Mexican and American physician-patient relationships; whereas American doctors defer the final decision to the patient, Mexican doctors are the decision-making authority, often giving directives to the patient.

Moreover, this locating of the decision-making power with the physician was often viewed positively (e.g., “like a parent”) while the reasons for American physicians’ orientation to patient autonomy was viewed somewhat negatively. When explaining this difference, Gabriela (52) stated: “Here [in the U.S.], they are more afraid to try things… in Mexico they are more open, more interactive…” Gabriela goes on to point out that Mexicans are much more personal and open and thus can be rightfully more authoritative. In contrast to this, the “boundaries” in American culture, as she says, “limit” American physicians from being properly authoritative. Others offered similar descriptions that characterized Mexican physicians as able to be properly authoritative and American physicians as being held back or limited by the nature of American culture.

Regardless of how these qualities are valued, the Mexican American respondents make a strong case for the cultural difference hypothesis in as much as they describe American physicians operating very differently from Mexican physicians. Considering that these respondents are the only group who have experience with both American AND Mexican physicians, this offers strong support for the cultural difference hypothesis. However, it also raises another important question: what might the White American respondents mean by “paternalism” when they described their physicians as being paternalistic?

##### White American Respondents’ Experiences with American Physicians

Interestingly, and in direct contrast to the Mexican American respondents’ relatively positive descriptions of paternalism, when describing their American physicians as paternalistic, White American respondents overwhelmingly described this paternalism in very negative terms. Six White American respondents described paternalistic physicians as being arrogant and nine others described paternalism in terms of dismissiveness. Seven White American respondents described a physician who does not listen to or respect the patient or as only asking superficial questions as an example of a paternalistic and thus problematic physician. The lack of these “care” factors were a common reason why many White American respondents felt that their American physicians were paternalistic, and in a very negative sense.

Irene (58), Emily (52), Matthew (54), and Alexa (48), each mentioned arrogance as a characteristic of paternalism. Irene shared that she feels that doctors are paternalistic due to being “arrogant”, because “they expect their patients to just do what they say and don’t give their patients the opportunity to ask questions”. Matthew offered a similar comment: “They [the doctors] kind of talk down to you…they send the message that ‘I’m the doctor, you’re the patient. Just do what I say.’” These physicians acting paternalistically were seen as arrogant and therefore negative by these American respondents.

Another term that many White American respondents used to describe the paternalism of American physicians was “dismissive”. This was used to describe situations where the doctor did not listen to the patient’s concerns, opinions, or even their symptoms. Alyssa (30), Megan (48), Susan (51), Amanda (34), and Patricia (38), among others, gave experiences or descriptions of their physicians being dismissive. For example, Susan said: “The doctor would not listen to concerns. I didn’t feel heard…valued…or trusted as somebody who knew symptoms… His word was the final word, and nothing else”. It should be clear here that dismissiveness is seen as one of the rather negative features of paternalism. Many of the other White American respondents mentioned that it is important that patients “feel heard” and “valued”. In describing one of her experiences of paternalistic physicians, Amanda shared her story about being diagnosed with Polycystic Ovary Syndrome (PCOS). In giving her diagnosis, this physician did not provide her with the opportunity to ask questions about her diagnosis or treatment options. She felt overwhelmed and confused, stating: “I was crying on the way home because I thought I was deathly ill until, like, I researched it more”. Here again, paternalism is being described in highly negative terms.

As we will see more clearly with the Mexican respondents, there are important differences in how the White American respondents describe paternalism as compared to how the Mexican and Mexican American respondents are describing it.

##### Mexican Respondents’ Experiences with Mexican Physicians

Similar to the descriptions from the Mexican American respondents, many Mexican respondents described their Mexican physicians as often telling the patient what to do or deciding for the patient what the treatment will be. Instead of consulting and letting the patient decide, the physician takes the main or sole role in making treatment decisions. Samantha (55), Victoria (38), Rosalia (52), Diego (33), Yanira (40), and others, each said that their physicians tell them what to do. Samantha’s description characterizes many of the Mexican respondent’s experiences well when she said: “In most cases, they tell me what to do, how to do it and when to do it… and in most cases they deliver”. Here, Samantha points to a clear paternalistic relationship in which the physician is highly directive and has authority over the treatment decision. Yanira shared a similar sentiment: “Yes, they tell you what to do. They say: ‘Do you want to be cured; you want to see a difference? Then you have to do this.’” Yanira points to the Mexican physicians as being highly directive. Yanira then further mentioned that she has confidence in and a preference for a paternalistic physician noting that if the patient doesn’t get better, it’s because they didn’t follow the doctor’s treatment decision. These experiences shared by the Mexican respondents clearly point to Mexican physicians as being paternalistic, while also indicating that these respondents appreciate their physicians’ paternalism and view it positively.

There were a number of different aspects of this positive evaluation of paternalism. For example, Angelica (37) pointed out these characteristics as she describes her physician as: “very professional, so he always talks to me with a lot of respect and tries to explain everything. Obviously, there are terms that I don’t understand because I’m not a doctor, but he tries to explain in simple terms my condition and what we’ll do”. As with the Mexican American descriptions of the paternalism of Mexican physicians, Mexican respondents often described their physicians’ paternalism in very positive terms such as “very professional” and as treating their patients with “respect” while still being the decision-maker as the physician tells the patient “what we’ll do”.

Other positive evaluations of their paternalistic physicians by Mexican respondents included a strong sense that their physicians really care about them. For example, Maria (55) shared: “Yes, the doctor is sometimes like a spiritual father, like a priest or bishop in the church…He also gives [personal] advice and sometimes you tell him personal things”. Another Mexican respondent, Sara, who described her physicians as paternalistic shared: “Well, we talk about other things not related to my problem that I have at that moment and that makes me feel like he sees me as a person, not just as a disease”. Samantha offered a similar sentiment about her physician who she described as paternalistic: “I have liked her [the physician] because…she is interested in all aspects, emotional and physical… She asks how you are, how’s your daughter, how’s work? She’s a doctor that’s very interested in her patients”. Furthermore, Diana shared: “In my personal experience, they have always been kind and respectful… [My doctor] always has a good attitude, he is looking me in the eye, he is attentive in the consultation”.

As we can see, Mexican respondents evaluated their paternalistic physicians very positively. These descriptions demonstrate that they see these physicians as “like a spiritual father”, caring, concerned, “attentive”, “very interested” in their patients, and as someone who makes their patients feel seen as a person. What is most striking about these Mexican respondents’ evaluations of their paternalistic physicians is that many of them are exactly opposite of the White American respondents’ evaluations of their paternalistic physicians. Where the White American respondents described their physicians as not caring, inattentive, and uninterested, the Mexican respondents described their physicians as caring, attentive, and very interested in them as a person.

In sum, the Mexican respondents overwhelmingly described their physicians as being paternalistic, and in ways that were described positively and appreciated by these patients. Regarding the seemingly contradictory nature of the experience data, we will revisit this below in the discussion.

## 4. Discussion

Considering the relatively small (20) and non-random nature of our respondents in each group, our research should be understood to be exploratory and our findings preliminary. With this caveat in mind, our findings provide strong support for the cultural difference hypothesis, both in terms of White American vs. Mexican respondents’ preferences for autonomy vs. paternalism (respectively) and in terms of differences of descriptions of experiences with Mexican vs. American physicians.

### 4.1. Differences in Preferences for Physician Paternalism vs. Patient Autonomy

Regarding the first, the White American respondents overwhelmingly said that they preferred a focus on patient autonomy with 60% choosing “Autonomy” and another 35% choosing “Mix” and just 5% choosing “Paternalism”. When White American respondents described what they meant by “Mix”, many of those who said “Mix” emphasized patient autonomy, making for a total of 17 out of 20 White American respondents who preferred “Autonomy”. In their explanations of this preference, they described a situation where the physician shares their knowledge with the patient but the patient ultimately makes the final decision regarding treatment. These respondents felt that this was the preferred arrangement because the patients know best about their bodies and life experience and because they have a right, even an obligation, of bodily autonomy.

In contrast, the majority of Mexican respondents chose “Mix” with “Paternalism” coming in as a close second and “Autonomy” third. Yet, when they explained their choices, those who said “Mix” emphasized paternalism as did three of the five who said “Autonomy”, making for a total of 15 out of 18 Mexican respondents who preferred “Paternalism”. In their explanations, they described a situation where the patient shares their knowledge with the physician but the physician ultimately makes the final decision regarding treatment. These respondents felt that this was the preferred arrangement because the physicians are the experts and have the authority, even the obligation, to make treatment decisions for their patients.

Furthermore, the logic used by White American respondents to justify their preferences for patient autonomy was almost a mirror image of the logic used by Mexican respondents to justify their preferences for physician paternalism. Whereas White American respondents felt that the final treatment decision should be made by the patient who is informed by the physician of the necessary medical knowledge, the Mexican respondents felt that the final treatment decision should be made by the physician who is informed by the patient of the necessary bodily knowledge. This suggests a different organization of what is morally valued in these two cultural contexts. As such, it provides strong evidence for the cultural difference hypothesis regarding preferences for autonomy vs. Paternalism.

### 4.2. Cultural Differences in Experiences with Physician Paternalism vs. Patient Autonomy

Regarding respondents’ experiences with autonomy and paternalism, responses to the question about whether their physicians were oriented towards autonomy or towards paternalism was, at first glance, less clear. Somewhat surprisingly given the findings in the literature on paternalism in American vs. Mexican healthcare, about the same number of White American and Mexican respondents described their physicians as paternalistic. At the same time, the Mexican American respondents contradicted the White American respondents’ characterizations of American physicians as paternalistic, indicating that they are in fact oriented to patient autonomy. This raises an important question: what can explain these differences in respondents’ characterizations of American physicians vis a vis paternalism?

A rather simple explanation might be to consider each group’s experiences. Whereas the White American respondents only have experience with American physicians (this was clear in their responses), the Mexican American respondents have had experiences with both American and Mexican physicians and thus can compare each relative to the other (something that was evident in their responses when they asked whether we were talking about American or Mexican physicians). This would suggest that the Mexican American respondents have more context by which to judge American physicians and that American physicians are generally oriented to autonomy and that the White American respondents were simply using “paternalism” as a negative term to characterize their physicians. This is a plausible interpretation of the data but we would like to suggest a more nuanced interpretation that might provide a richer understanding of the ways in which culture is relevant to the question of autonomy vs. paternalism.

It may be that both the Mexican American and the White American respondents’ characterizations of American physicians are accurate when considered relative to their own cultural frameworks. We can see two ways in which this cultural relativism works: first in how paternalism is defined and second in what it means when a physician acts paternalistically.

Regarding the first (i.e., what paternalism means), having seen the paternalism of Mexican physicians and with an understanding of the positive value of paternalism in that context, Mexican American respondents see American physicians as oriented to patient autonomy (in ways that are often seen negatively and thus in need of an excuse or explanation). On the other hand, White American respondents only have known American physicians and their own ideals of how a physician should act. Here, the ruling ideal is the autonomy of the patient and thus when White American respondents consider their physicians they find that their physicians do not compare well to their ideal of patient autonomy and conclude that their physicians are paternalistic. Thus, each group’s characterization of American physicians vis a vis paternalism is relative to their experiences and cultural understandings of and ideals about what counts as paternalism, which appears to differ between these two groups.

Regarding the second (i.e., what it means for a physician to be paternalistic), we can see how the meaning of paternalism is evaluated differently in these two cultural contexts. Whereas the Mexican respondents consider their paternalistic physicians to be caring and concerned and very interested in them as a person, the White American respondents consider their paternalistic physicians who make decisions for them to be exactly the opposite—uncaring, unconcerned, and uninterested in them as a person and of their knowledge of their own bodies and of their bodily autonomy. This points to an important difference in the very moral evaluation of what it means for a physician to be paternalistic.

Both of these features point to more complex ways in which cultural differences function as fundamental organizing principles. This definitional question needs to be taken seriously by researchers. What is needed here is a rich understanding of what paternalism looks like in actual physicians’ practices (e.g., Thompson and Whiffen 2018). The second concern of cultural relativism, what paternalism means, points to cultural differences in what it means to a patient to have an encounter with a paternalistic physician. Again, here it will be essential to observe patients’ experiences with paternalistic physicians in order to explore and better understand those patients’ reactions and characterizations of the encounter as well as the patients’ consequent behaviors (esp. regarding things like treatment adherence and trust in their physician).

This also suggests that we consider paternalism in light of moral pluralism [19]. It should be clear that if we assume the White American respondents’ characterizations of physician paternalism as uncaring, arrogant, disrespectful, dismissive, uninterested in the patient, and so on, then paternalism in a physician-patient relationship would in no way be a part of high-quality patient care. Yet, if we assume the Mexican respondents’ characterizations of physician paternalism as caring, professional, respectful, “very interested” in their patients’ lives, and so on, it would seem that physician paternalism is a necessary part of high-quality patient care. This again points to the importance of understanding the cultural contexts in which paternalism is defined and interpreted so that we can better understand the consequences that cultural differences will have for patients.

We would also like to note here that these findings raise important questions regarding why these Mexican respondents appear to have so much more positive relationships with their physicians as compared to the American respondents. These descriptions were particularly striking in light of the very negative, almost exactly opposite, descriptions that American respondents gave of American physicians. Although this quickly moves beyond the scope of the present study, our findings here do suggest that the paternalism of Mexican physicians demonstrates, for Mexican respondents, that Mexican physicians are highly caring, interested, and invested in their patients as individuals—a stark contrast with White American respondents’ evaluations of their American physicians as uncaring, dismissive, and uninterested in them as patients.

We would further note that healthcare systems and institutions themselves contribute to the culture of healthcare in a given place. Thus there are likely other aspects of American healthcare systems and institutions, such as the proliferation of “defensive medicine” in response to a strong culture of litigation, that contribute to the emphasis on patient autonomy in the U.S. Importantly, this points to the fact that culture is not a simple matter of individual preferences or simply of good or bad physicians. Rather, cultural differences in physicians’ practices are the result of a complex interaction between encultured individuals and encultured systems and institutions.

## 5. Conclusions

Based on our interviews with sixty (60) Mexican, Mexican American, and White American respondents, we found overall strong support for the cultural difference hypothesis with regard to preferences for and experiences of patient autonomy vs. physician paternalism. Whereas White American respondents preferred physicians oriented to patient autonomy, Mexican respondents preferred paternalistic physicians. With regard to their experiences, both White American and Mexican respondents tended to describe their physicians as paternalistic, only whereas the Mexican respondents described their physicians’ paternalism in overwhelmingly positive terms, the White American respondents described their physicians’ paternalism in overwhelmingly negative terms. Importantly, the Mexican American respondents, most of whom had experience with physicians in both places, indicated that whereas their American physicians were oriented towards autonomy, their Mexican physicians were oriented towards paternalism. This was in contrast to the White American respondents who tended to see their American physicians as paternalistic. As just outlined in our discussion, this provides further support for the cultural difference hypothesis by pointing to cultural differences in what paternalism means and what it means for a physician to be paternalistic.

The cultural difference hypothesis has some potentially important implications for healthcare practice. Two particularly important aspects that two of us previously proposed are demonstrations of physician care for patients and patients’ treatment adherence (Thompson and Whiffen, 2018).

Regarding “care”, our current study strongly suggests that physician paternalism is important for patients’ perceptions of their physicians’ care for them. Our data further suggest that these perceptions are determined relative to patients’ cultural backgrounds. Whereas our White American respondents viewed paternalism as a deeply problematic physician behavior, our Mexican respondents viewed paternalism as a way that their physicians demonstrate their care for them as patients, as an indication of their interest in them as individuals. An understanding of how paternalism might be differently understood in different cultural contexts could have important implications for physicians interested in demonstrating care for and building trust with their patients.

Regarding treatment adherence, there was evidence pointing to the possibility that this cultural difference between Mexican patients and American physicians regarding how physicians interact with patients might result in lower levels of treatment adherence among these patients. In particular, the Mexican respondents repeatedly referred to paternalistic Mexican physicians in positive ways that indicated that the patients had confidence in the abilities of their Mexican doctors and that they saw them as trustworthy authorities that were supposed to be very directive when telling their patients what they needed to do. The Mexican American respondents also described what they saw as their American physicians’ orientation to patient autonomy in negative terms that were in need of an explanation—as if they *should* be more directive but they cannot be more directive because it does not fit with the culture. This suggests that Mexican patients who are given treatment options by an autonomy-oriented physician might have less confidence in the authority of their physicians and thus be less likely to follow through with the treatment.

Minimally, both of these possibilities suggested by this and other research provide a strong argument for the importance of better understanding the role that cultural differences might play in healthcare practices. Our research here suggests that understanding these kinds of cultural differences is likely to be essential to improving treatment adherence while also creating trusting and caring physician-patient encounters.

## Figures and Tables

**Figure 1 ijerph-19-10663-f001:**
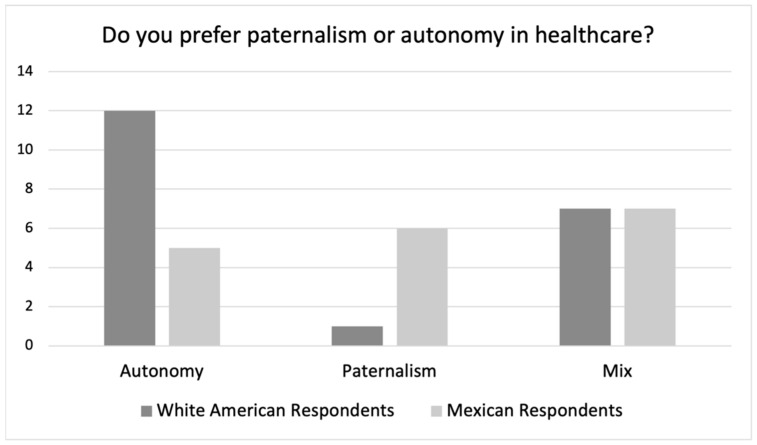
White American and Mexican Respondents’ Preferences.

**Figure 2 ijerph-19-10663-f002:**
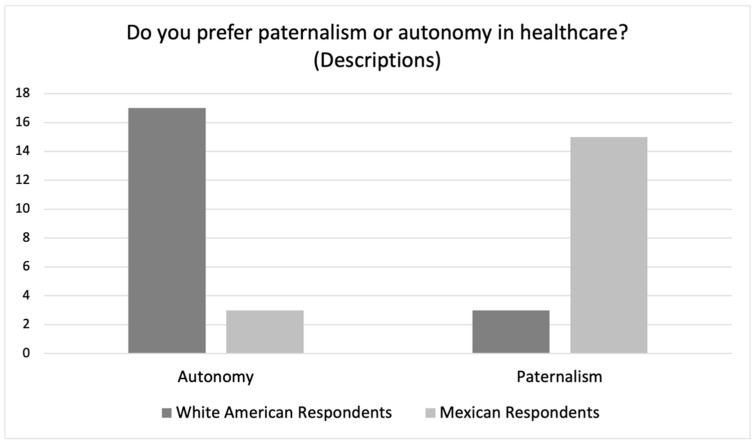
White American and Mexican Respondents’ Preferences Based on Explanations.

**Figure 3 ijerph-19-10663-f003:**
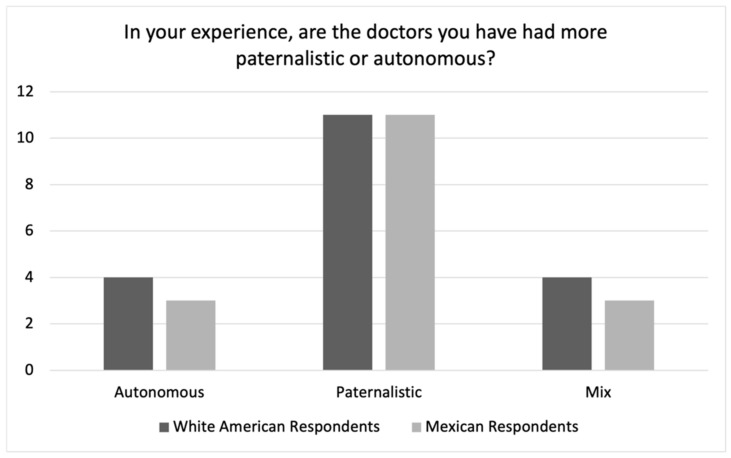
White American vs. Mexican Respondent Experiences.

**Figure 4 ijerph-19-10663-f004:**
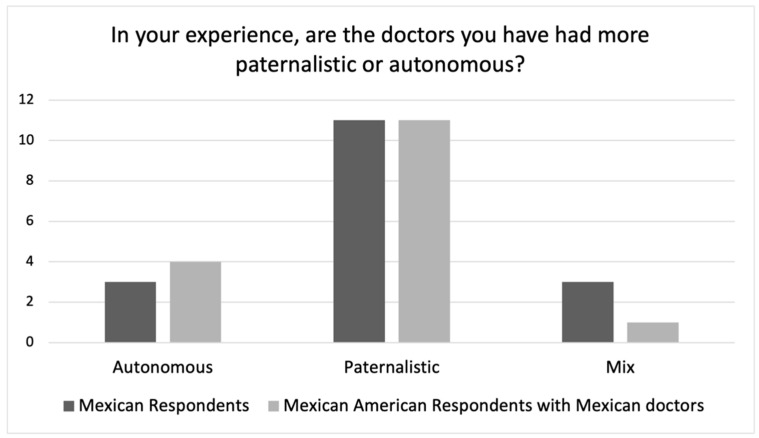
Mexican American vs. Mexican Respondent Experiences.

**Figure 5 ijerph-19-10663-f005:**
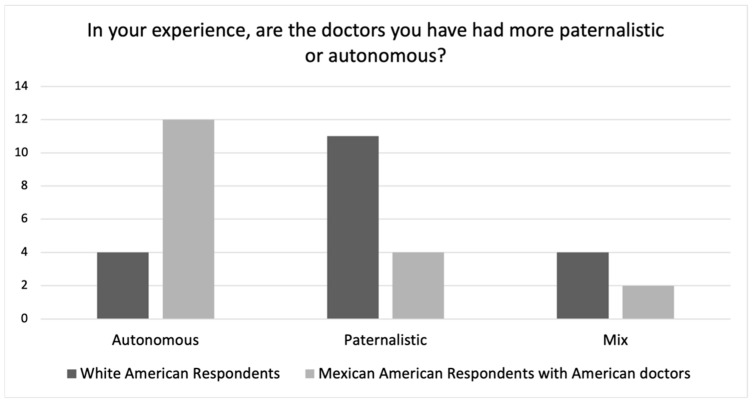
Mexican American vs. White American Respondents’ Experiences.

**Table 1 ijerph-19-10663-t001:** Origin of Participants for the Three Groups.

White American		Mexican American		Mexican	
Location	Number	Location	Number	Location	Number
California	3	California	2	Hidalgo	16
Idaho	1	Kansas	2	Nuevo Leon	1
Texas	1	New Mexico	2	Tamaulipas	2
Utah	10	Texas	2	Veracruz	1
Washington	5	Utah	5		
		Washington	7		
**Total**	20		20		20

**Table 2 ijerph-19-10663-t002:** Participant Information.

	Interview Language		Gender		Age		Interview Time
Group	Spanish	English	Female	Male	Range	Mean	Average (min:sec)
White American	0	20	12	8	30–58	47.5	36:39
Mexican American	3	17	15	5	31–60	48.6	39:27
Mexican	20	0	15	5	33–65	48.7	30:42
**Total**	23	37	42	18	30–65	48.2	35:36

## Data Availability

The data presented in this study are available on request from the corresponding author pending IRB approval which must be submitted and completed by the requesting researcher. The data are not publicly available due to privacy and possible immigration information discussed in the IRB.

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
