# Peer review of "Cultural Differences in Patients’ Preferences for Paternalism: Comparing Mexican and American Patients’ Preferences for and Experiences with Physician Paternalism and Patient Autonomy"

_ijerph, 2022, doi:10.3390/ijerph191710663_

Round 1
Reviewer 1 Report
1. The concepts of paternalism and automy may differ depending on the background of the subject. But the interview question was “Do you prefer paternalism or autonomy in healthcare?” . This is judged to be incorrect in the structure of the article, and the researcher first defines and explains “the concept of paternalism and automy in our study” and then “Do you prefer paternalism or autonomy in healthcare?” I think the order is to ask.
Although there is a concept in the research method, there is no procedure in Appendix A to define “paternalism and automy”and inform the research subjects. If there is such a procedure, please include it in the research method, and describe in detail which question in the interview followed by this information.
2. Appendix B is judged to be the important research result, but why is it only included in Appendix B and not described in the research result?
3. This study is a qualitative study, and the main results of the study only contain Yes or No answers to the interview questions in Figures 1 to 5. Does these figures can show “cultural differences in preferences for and experiences with physician paternalism vs. patient autonomy in mainstream U.S. Can the figures show me “culture as compared with Mexican culture”? I don't understand the “cultural differences” of the present state of the findings, nor why a qualitative study was necessary. All will need correction and rearrangement.
Author Response
We are grateful to Reviewer 1 for their comments and suggestions and have done our best to address them as follows:
- R1 raises asks us to clarify whether or not the definitions of autonomy and paternalism preceded the question about whether they prefer autonomy and paternalism. On page 5, line 172, we clarified the ordering with the following sentence: “Before asking any questions about paternalism and autonomy, we provided the following simple definitions of autonomy and paternalism:”
- In their second comment, R1 says that “Appendix B is judged to be the important research result” even though nowhere in the text do we say this. The only reference to Appendix B (or C) are at the beginning of the methods section. Because these are not critical to our analysis, we wondered if they might be distracting and thus we considered removing them. In the end, we felt that we need to include them in the interests of being as transparent as possible about our method of data analysis. On the other hand, we feel strongly that bringing those codebooks into the text would only confuse the reader by overwhelming them with details that are not necessary for following the logic of and evidence for the argument we are making in this article. If the editors feel that one or the other steps need to be taken, we can do that. But we feel comfortable with the presentation of the methods and data in our revised draft.
- In their third comment R1 suggests that Yes or No answers aren’t sufficient for answering questions about cultural differences. We agree entirely with R1’s point here, and this is why although the article starts with quantitative data, the focus is on the qualitative data that provides much more insight than the Yes/No responses. We make this structure of the logic of the article’s argumentation clear in the following sentences in the paper:
“We begin with the quantitative data regarding respondents’ preferences for patient autonomy vs. physician paternalism. We then explore these data further by describing our respondents’ explanations of their answers to this question. In 3.2, we turn to respondents’ answers to the question of actual experiences with physicians, again beginning with the quantitative data and then adding nuance and detail with the qualitative data describing their explanations for their answers.” (p. 5, lines 208-213).
As to the claim that qualitative research cannot show “culture” or “cultural differences,” we are confused. As researchers trained in anthropology, we would argue that qualitative research is in fact the best way to document and understand cultural differences. Perhaps this is a difference of disciplinary training or some other difference in an understanding of methodological and epistemological constraints.
Reviewer 2 Report
Thank you for the opportunity to review this manuscript on the culturally distinct preferences and experiences of paternalism and autonomy in health care.
I found the manuscript to be very well-written, thoughtfully structured, and easy to follow. I also think that the manuscript makes an important contribution to the critical engagement with taken-for-granted assumptions around paternalism and autonomy beyond Western biomedical bioethics. I have some suggestions to further strengthen the manuscript, mostly related to the study design and the way that qualitative data is conceptualised and used. Please see details below:
1. The manuscript elaborates to some extent how the authors established the categories of ‘mainstream American’, ‘Mexican-American’ and ‘Mexican’. I still think that these categorisations are somewhat problematic, especially in a racialized social and health system like the USA. What exactly is a mainstream American? If the authors mean to capture a White/European ancestry this should be made explicit. If this category captures all other categories other than ‘Mexican’ this should also be made clear. I also would like to see a note on who assigned participants in the sample to these categories, for example did the researchers make this judgment based on some criteria during the recruitment process or is this based on self-reported identification of participants?
2. The Introduction provides a compelling rationale for why the prominence of autonomy needs to be interrogated within socio-cultural context by outlining how community-centred culture might engender preferences for more paternalistic approaches. The authors review literature examples drawn from many cultural and national contexts. It may be worth adding here potentially, that also in Western bioethics patient autonomy has been interrogated critically and consistently. I was reminded here of work by Glaser & Strauss (1979) ‘Awareness of dying’ and Timmermans (1994) ‘Dying of awareness’. Anecdotally, it has come up in my own research practice that many people from White European background also are not that clear on their preferences for autonomy.
3. Related to that, I was wondering if the context of health care could be specified for the reader, for example, were participants primarily talking about primary health care, or generally about encounters with any type of physician. I’m asking this, because there may be the severity of the health issue to factor in. When people are presenting with potentially life-limiting illness their preferences for autonomy may shift dramatically regardless of the cultural background. If this is beyond the scope of the present manuscript, I suggest clarifying the context in which this study is interested, and perhaps add a potential for further research to follow-up in the discussion.
4. The study design seems to rely on interviews with 60 participants from 3 distinct participant groups. The qualitative design sits uneasily with the formulation of a ‘cultural difference hypothesis’ that is tested in the analysis. I also do not think that this dataset can be treated as both quantitative and qualitative, including Chi-squares and frequency counts. First, despite some attempts at acknowledging a non-random sample as a ‘caveat’, interview data simply cannot be treated as if it was a survey. Second, I don’t think it adds value to the manuscript either. There is a strong and persuasive narrative that the authors tease out of the interview data and its interpretation. All the quantitative treatment of the interview data does for me is to raise issues around validity and rigour.
5. On page 11, paragraph one I disagree with the logic the authors are proposing. I believe what they are capturing is that there are differences between preferences and experiences, which they expertly tease out in later parts of the analysis. I wonder, then why again the attempt to make sense of the data by means of the cultural difference hypothesis?
6. On page 12, paragraph two. How sure can the authors be that only Mexican-American participants had experiences with both groups of health care providers? Is this indeed what explains their nuanced discussions of autonomy and paternalism with either health care provider?
7. The quote on page 13, first paragraph seems to allude to the practice of ‘defensive medicine’ and litigation culture in the USA. It would be nice to make that explicit (if indeed this is the case). This would help to further situate and specify autonomy and paternalism within a larger cultural system beyond individual doctor’s practices.
8. Page 15, lines 631-634; Qualitative data analysis is really not raising the question ‘who got it right?’ and the authors quickly move on to propose a more productive way to engage with their findings in the remainder of the discussion. I suggest deleting this sentence.
9. There are some inconsistencies, duplication and missing information in the reference list.
Despite these suggestions, I would like to reiterate that the study presents important and relevant findings in an engaging way. It is hoped that any revisions (if the authors agree with the suggestions) will be further improving the manuscript.
Author Response
We are grateful to Reviewer 2 for their very thoughtful comments and suggestions and have done our best to address them as follows:
- R2 notes the need to specify the selection criteria for specifying “mainstream American”. We agree with this concern. Given how we selected participants, we have realized that it would be more accurate and proper to use the term “White American” rather than “mainstream American”. We have added that throughout the document. We have also included the following sentences to clarify our selection criteria: “White Americans were U.S. citizens who self-identified as “White”. Mexican American participants were born either in the U.S. and have parents or grandparents from Mexico or or were born in Mexico. All Mexican American participants currently reside in the U.S. but citizenship status remained unidentified in case revealing that status might lead to harm (e.g., deportation).”
We also rewrote the footnote noting that the terms “White American” and “Mexican” are somewhat problematic. We have left this in a footnote because it is an issue that will be of serious to concern to anthropologists but will likely not be of much concern for most readers of IJERPH.
- R2’s second comment notes that there have been cultural-internal (to Western culture) criticisms of autonomy in bioethics. Although this does exceed the scope of this article’s focus on cross-cultural critiques, we have added the following two sentences to note that there are cultural-internal critiques: “Indeed, even some research from within Western contexts has suggested similar limits on the ethic of patient autonomy. Glaser and Strauss (1965) and Timmermans (1994) have separately demonstrated how the autonomy of dying patients can be limited by the awareness that is afforded them of their condition by others” (p. 3).
- In their 3rd comment R2 notes the need to specify if any healthcare contexts were specified in the interview process. We agree with this suggestion and have included a sentence in the methods describing that we left this open ended such that our respondents made reference to both primary care providers and specialists of various sorts, sometimes including more serious illnesses (“With these and other questions, we did not specify the context of the kind of physician. Rather, we left this up to the participants to define. Most talked about their primary care providers but some mentioned specialists of various kinds”). We considered but ended up not adding a “further research” point related to this since it felt like it was beyond the scope of this study – i.e., there is no obvious reason why more severe contexts of care would impact our findings regarding cultural differences. We would expect both groups to shift to favoring paternalism more. (and, of course, we could be wrong, but nothing in our data would suggest a need to explore this particular angle further).
- R2’s fourth comment raises the question of whether the cultural difference hypothesis can be adequately tested with qualitative research. We feel that this is a misinterpretation of the intentions of our article. When we introduce the cultural difference hypothesis, we do not see our project as “hypothesis-testing” strictu sensu. Our aim is not to “prove” the hypothesis right or wrong. Instead, we consider our article to be “hypothesis exploring”, that is to say, we are exploring our data in light of the hypothesis that there might be cultural differences in the experiences and preferences with healthcare interactions between Mexican and mainstream Americans. We feel that our language throughout the article is fairly clear in this regard (but please point us to language that is stated too categorically). Nonetheless, in order to make this entirely clear, we have added the following at the very beginning of the Results section:
“On other caveat is in order. To be sure, our study is not intended as an hypothesis-testing study (which would require a more rigorous selection methodology and more rigorous statistical methods of analysis). Rather, our study is intended as an hypothesis-exploring study in as much as we explore the cultural difference hypothesis in light of our participants’ responses. Our findings should be considered as preliminary findings and, we hope, serve as the basis for future hypothesis-testing research. But we also feel that these data are quite compelling as they are.” (p. 6).
To this point, we believe that the statistical data such as the chi squared analysis helps us explore the hypothesis and we decided to keep it. If Reviewer2 or the Editors feel strongly that it is too distracting or confusing even with the above clarification, we are willing to delete it.
- Here R2 notes that they disagree with the logic of the article regarding the cultural difference hypothesis. Similar to the previous comment, we hope that our clarification emphasizing the hypothesis-exploring nature of our study will be helpful. Rather than proving a point, we are supplying data and a discussion that explores the hypothesis. To this effect, we consider data that both shows support for and contradicts the cultural difference hypothesis regardless of whether or not it does so in anything approaching a definitive manner. As such, we do not see it logically problematic to note that some of the data contradict the hypothesis as stated.
- In their sixth point, R2 notes the need to specify if we know that only Mexican-American participants had experiences with both Mexican and American physicians. As seen in some of our interview questions in Appendix A (questions 3, 7, 8, 9, 10, 22), we addressed the possibility that respondents from any of the groups would have had experiences with physicians from the other culture. Many of our Mexican-American respondents had experiences with both American and Mexican physicians as they would ask frequently which of the doctors we were asking about. We did not have similar questions from the American respondents. We did have a few Mexican respondents who had experiences with American physicians when they were visiting family members in the U.S. and their comments were in line with the Mexican-American respondents.Yet since there were only a few who had these experiences so we did not focus on this since the Mexican-American data was much clearer. We added two parentheticals to address this on p. 16:
“Whereas the White American respondents only have experience with American physicians (this was clearl in their responses), the Mexican American respondents have had experiences with both American and Mexican physicians and thus can compare each relative to the other (this was evident in their responses when they asked whether we were asking about American or Mexican physicians).”
- In their seventh point, R2 notes that one of the comments (Gabriela p. 13) seems to allude to the practice of defensive medicine and litigation culture in the U.S. and that this should be made explicit. In looking more closely at the quote, it was clear that this quote was about the closeness of Mexican physicians as opposed to the distanced American physicians. As a result of this, she was saying that since American physicians don’t know their patients as well, there isn’t the same sense of trust and they have to be more cautious.
At the same time, we agree with the point of R2’s suggestion and have added the following language on p. 17:
“We would further note that healthcare systems and institutions themselves contribute to the culture of healthcare in a given place. Thus there are likely other aspects of U.S. healthcare systems and institutions, such as the encouragement of “defensive medicine” in response to a strong culture of litigation, that contribute to the emphasis on patient autonomy in the U.S. (since involving patients in decision-making takes some of the legal responsibility off the physician). Importantly, this points to the fact that culture is not a simple matter of individual preferences or of good or bad physicians. Rather, cultural differences in physicians practices are the result of a complex interaction between encultured individuals and encultured systems and institutions.”
- Here R2 notes that qualitative data analysis is not raising the question of who got it right. Again, we feel here that R2 is reading the logic of the article as hypothesis-testing. We hope that the additions mentioned in point #4 above will address this. But we also agree that the question could be worded more accurately (perhaps if not quite so cutely) as follows: “This raises an important question: what can explain these differences in respondents’ characterizations of American physicians vis a vis paternalism?” Other language throughout this section has been changed to make it clear that we are exploring our respondents’ answers.
We would further note that, in agreement with R2’s concern here, in our evaluation of this response, we do not use the data to make the case that one or the other group is right. Rather, in the interests of exploring the proposed hypothesis, we offer some possible explanations for what might be going on here. In the end we suggest that both are correct – a conclusion that further points to a more elaborate difference in culture, namely that the very definitions of the terms are culturally contingent.
We have gone through to ensure that the sources and bibliography match and have removed duplicates.